# Quantitative Characterization of Motor Control during Gait in Dravet Syndrome Using Wearable Sensors: A Preliminary Study

**DOI:** 10.3390/s22062140

**Published:** 2022-03-10

**Authors:** Maria Cristina Bisi, Roberto Di Marco, Francesca Ragona, Francesca Darra, Marilena Vecchi, Stefano Masiero, Alessandra Del Felice, Rita Stagni

**Affiliations:** 1Department of Electrical, Electronic and Information Engineering “Guglielmo Marconi”, University of Bologna, Viale del Risorgimento, 2, 40136 Bologna, Italy; mariacristina.bisi@unibo.it (M.C.B.); rita.stagni@unibo.it (R.S.); 2Health Sciences and Technologies-Interdepartmental Center for Industrial Research, Via Tolara di Sopra, 50, Ozzano dell’Emilia, 40064 Bologna, Italy; 3Department of Neuroscienc, University of Padova, Via Belzoni 160, 35121 Padova, Italy; roberto.dimarco@unipd.it (R.D.M.); stef.masiero@unipd.it (S.M.); 4Department of Paediatric Neuroscience, Euroepan Reference Network EpiCARE, Fondazione IRCCS Istituto Neurologico Carlo Besta, Via Celoria, 11, 20133 Milano, Italy; francesca.ragona@istituto-besta.it; 5Pediatric Neurology, University Hospital of Verona, P.Le Stefani, 1, 37121 Verona, Italy; francesca.darra@univr.it; 6Department of Women and Children Health, University of Padova, Via Nicolò Giustiniani, 3, 35128 Padova, Italy; marilena.vecchi@unipd.it; 7Padova Neuroscience Centre, University of Padova, Via Giuseppe Orus, 2, 35131 Padova, Italy

**Keywords:** gait analysis, SCNA1 mutation, IMUs, motor control, entropy, recurrence

## Abstract

Dravet syndrome (DS) is a rare and severe form of genetic epilepsy characterized by cognitive and behavioural impairments and progressive gait deterioration. The characterization of gait parameters in DS needs efficient, non-invasive quantification. The aim of the present study is to apply nonlinear indexes calculated from inertial measurements to describe the dynamics of DS gait. Twenty participants (7 M, age 9–33 years) diagnosed with DS were enrolled. Three wearable inertial measurement units (OPAL, Apdm, Portland, OR, USA; Miniwave, Cometa s.r.l., Italy) were attached to the lower back and ankles and 3D acceleration and angular velocity were acquired while participants walked back and forth along a straight path. Segmental kinematics were acquired by means of stereophotogrammetry (SMART, BTS). Community functioning data were collected using the functional independence measure (FIM). Mean velocity and step width were calculated from stereophotogrammetric data; fundamental frequency, harmonic ratio, recurrence quantification analysis, and multiscale entropy (τ = 1...6) indexes along anteroposterior (AP), mediolateral (ML), and vertical (V) axes were calculated from trunk acceleration. Results were compared to a reference age-matched control group (112 subjects, 6–25 years old). All nonlinear indexes show a disruption of the cyclic pattern of the centre of mass in the sagittal plane, quantitatively supporting the clinical observation of ataxic gait. Indexes in the ML direction were less altered, suggesting the efficacy of the compensatory strategy (widening the base of support). Nonlinear indexes correlated significantly with functional scores (i.e., FIM and speed), confirming their effectiveness in capturing clinically meaningful biomarkers of gait.

## 1. Introduction

Dravet syndrome (DS) is a severe childhood-onset epilepsy syndrome, related to a genetic mutation of the sodium channel alpha-1 subunit [1]. It affects between 1/15,000 and 1/40,000 individuals [2]; DS children develop frequent and pharmacoresistant polymorphic seizures. Besides cognitive stagnation, people with DS develop neurological signs—mainly represented by ataxia, pyramidal signs, and myoclonus—and gait alterations, resulting in progressive severe gait deterioration [3,4], leading to unstable walking and a high risk of falls. Consequently, people with DS have a lower grade of independency and they and their caregivers experience a poor quality of life.

Ataxia is a neurological sign of motor dyscontrol. Cerebellar ataxia is related to a dysfunction of the cerebellum, a posterior portion of the brain principally involved in the feedback/feedforward adjustment of motor tasks, with a specific impairment in synchronization of the coordination of muscle contractions. Among motor tasks, trunk control is also affected, contributing—together with limb incoordination—to gait instability in cerebellar ataxia. Other signs of cerebellar ataxia include slurred speech and abnormal eye movements. A clinical picture which resembles cerebellar gait instability may be instead related to decreased sensation in the lower limbs, namely decreased proprioception, which has clinical distinguishing features and presents with a milder clinical picture.

Among DS-related gait alterations, crouch gait, a pattern typically observed in cerebral palsy and characterized by hip and knee flexion and femur anteversion, was originally identified via observational video gait analysis [5]. Quantitative gait analysis demonstrated that only a subgroup of children and young adults with DS display gait pattern with similar, but not equal, characteristics of crouch gait, termed pseudo-crouch gait [6,7]. The increased flexion of lower limb joints is likely related to stabilization strategies in a clinical picture often characterized by reduced muscle tone, cerebellar ataxia, and moderate hyposthenia. Instrumental gait analysis also permitted detection of higher muscular efforts and energy expenditures in the same population, with compensatory strategies to promote propulsion (i.e., forward lean of the trunk in younger subjects) [7]. A recent report [8] aimed to study foot function of people with DS using pedobarography-confirmed features of walking instability and motor development immaturity that normally disappear with growth in healthy subjects [9,10]. 

Gait pattern characterization in DS can play a fundamental role in the prognosis of the pathology [11]. In fact, it remains challenging due to the variability of the observed alterations and to the scarce collaboration of DS subjects, due to intellectual disability and behavioural disturbances [3,6]. Less invasive quantitative assessment techniques, easier to apply in ecological conditions, can better suit the study of DS gait, aiming to provide insight into the mechanisms underlying the alteration of motor control (e.g., immaturity, instability) and to potentially identify prognostic biomarkers related to the aforementioned control alterations to guide rehabilitative interventions.

The limited invasiveness of wearable inertial sensors can significantly simplify the routine assessment of DS subjects. Human movement analysis methods, based on wearable inertial sensors, have been widely adopted to study human motion during static and dynamic conditions in different pathologies [12,13,14,15,16,17]. These have for example been used in identifying parkinsonian and ataxic features [18,19,20,21,22] allowing the quantitative assessment in outpatient settings throughout the life span, effectively integrating the information derived from qualitative observation. In particular, nonlinear indexes calculated from IMU-acquired gait data allowed the quantitative characterization of motor control characteristics (i.e., complexity and automaticity) in infants, toddlers, and young adults with typical and atypical development [23,24,25]. Nonlinear indices, such as multiscale entropy [26] and recurrence quantification analysis, allowed researchers to quantitatively assess locomotor maturation during the life span [27,28], highlighting differences related to age maturation [28] and providing information complementary to standard clinical tests [29,30]. Therefore, such metrics retain the potential to quantitatively characterize DS immature gait pattern, simultaneously considering the potential effect of age.

We hypothesized that the selected nonlinear metrics can quantify the characteristics of motor control alteration in a cohort of DS subjects: we expected that they may highlight an immaturity of motor control in terms of instability, signalled by reduced motor complexity, automaticity, and stability (i.e., reduced indices for multiscale entropy and recurrence quantification analysis) as compared with the typical development control group. These metrics may provide significant additional objective information on the characteristics of motor control in DS. The aim of the present work was to analyse the gait of individuals with DS over a large age span, using a sensor-based approach to characterise the acceleration at the lower back, which is related to the control of the progression of the centre of mass [28,29], to motor control, and to the age of maturation.

For this purpose, the gait performance of a cohort of DS participants was first compared to reference data for gait development in the same age range. Subsequently, we compared the DS metrics to different specific age groups to evaluate the DS stage of motor control development with respect to typical motor control maturation. This analysis can contribute to deepening our knowledge of the mechanisms underlying the alteration of motor control in DS subjects, with particular attention to the maturation of gait. This approach can eventually become part of the routine monitoring of these subjects as potential prognostic biomarkers.

## 2. Materials and Methods

### 2.1. Settings and Ethics Statement

DS subjects were recruited at the Teaching Hospital of Padova (Italy), the Verona University Hospital (Italy), and the Neurological Institute Carlo Besta (Milan, Italy). Data collection was performed from March 2018 to July 2020 at the Laboratory of Clinical Analysis and Biomechanics of Movement and Posture of the Teaching Hospital of Padova, Italy. Ethical approval was granted by the local board in Padova (protocol number 4276/AO/17). Typical development (TD) control subjects were recruited from local Italian primary and secondary schools (Istituto San Giuseppe and High School in Lugo (RA)), and among students at the University of Bologna. Ethical approval for the control group was granted by the Bioethical Committee of the University of Bologna (25 May 2016). Subjects, or their legal guardians, provided written informed consent to participate in the study.

### 2.2. Inclusion and Exclusion Criteria

Individuals with genetic diagnosis of DS were eligible to participate. Inability to walk independently or reported seizures within 24 h prior to the data collection session were exclusion criteria. Participants were included if aged 5 years or older, to ensure that potentially identified gait abnormalities were related to underlying neurological/biomechanical issues and were not deviations from physiological developmental milestones.

The control sample included children from 6 to 15 years of age and 25-year-old young adults. All children were born at full term, with no orthopaedic (i.e., flat foot, spinal deformities) or neurological/developmental disorders.

### 2.3. Subjects

A total of 20 subjects (7 males, aged between 9 and 33 years) with DS were included.

A total of 112 TD control subjects were divided into 7 groups of 16 subjects each (8 females and 8 males), based on their age, and included in the study. Control group data were presented in a paper by Bisi et al., 2019 [28].

All the participants, DS and TD, had a body mass index between the 5th and the 95th percentile of the body mass index for age [31]. Characteristics of DS and TD subjects are reported in Table 1 and Table 2, respectively.

All DS participants had an abnormal neurological examination, with pyramidal signs in 9, extrapyramidal signs in 3, upper limb dysmetria in 13, limb myoclonus in 2, and trunk ataxia in 15.

### 2.4. Data Collection

In DS subjects, community functioning data were collected using the functional independence measure (FIM) [32]. This scale is an 18-item, 7-level ordinal scale instrument used to assess adult and children performance in self-care, sphincter control, transfers, locomotion, communication, and social cognition. The score ranges between 18 (the lowest independence and the highest need of assistance) and 126 (the highest level of independence, no assistance needed). The 18 items are grouped into 3 domains: activities of daily living (8 items; subscore: 8–56), motor function (5 items; subscore: 5–35), and cognition (5 items; subscore: 5–35).

Three tri-axial wireless inertial sensors (OPAL, Apdm, Portland, OR, USA; Miniwave, Cometa s.r.l., Italy) were mounted on the lower back (L5 level) and on the shanks (above lateral malleolus) of DS and TD subjects using straps (Figure 1).

Data were sampled at 128 Hz while the subjects walked at self-selected speed back and forth along a straight path. Tests were performed in the motion analysis laboratory of the Teaching Hospital of Padova (Italy) for DS participants and in schools/universities for the control group [28,30].

In DS participants, the walking trials were also acquired by means of stereophotogrammetry (10 cameras, SMART D-500, BTS Motion Capture, Italy–200 Hz) based on the Davis protocol [33].

### 2.5. Data Processing

In DS, walking speed (m/s), normalized walking speed (normalized by the subject height, H, %H/s), and step width (cm) were calculated from stereophotogrammetric data. Walking speed (m/s) was defined as the average speed of the markers placed on the heels between foot-strike events over the whole walking trials. Step width (cm) was defined as the distance in the medial–lateral direction between the left and right heel markers at two subsequent foot-strike events.

The turns and the first and the last two strides of each walking section were removed from the wearable inertial sensor signals before further analysis [34]. For all participants, 14 strides were analysed, being the maximum number of strides identified. The number of strides was identified from the angular velocity around the mediolateral axis of the shanks [35].

The following indices [36] were calculated from trunk acceleration data:-Fundamental frequency (FF, in Hz), calculated as the maximum of the spectral distribution of the lower back acceleration signal; in healthy mature gait, the distinctive peak is related to cadence [37].-Harmonic ratio (HR), related to rhythmicity, was calculated on trunk acceleration data along the 3 directions (vertical—V; mediolateral—ML; anteroposterior—AP), decomposing the signal components into its harmonics, as the ratio between the sum of the first 10 even and the first 10 odd harmonic multiples of the FF [34,38].-Recurrence quantification analysis (RQA), related to automaticity and pattern regularity, implied the calculation of recurrence rate (RR), determinism (DET), and averaged diagonal line length (AvgL) for each acceleration component (V, ML, and AP) [30,39].-Multiscale entropy (MSE), related to complexity and automaticity, was calculated as the sample entropy (SEN) of trunk acceleration components (SENv, SENml, SENap) at time scales (τ) from 1 to 6: (i) coarse-grained time series were calculated by averaging the increasing numbers of the data points in non-overlapping windows of length, τ, τ = 1:6; (ii) length of sequences to be compared, m, was fixed at 2, and tolerance for accepting matches, the radius, was fixed at 0.2 [30].

HR was selected given its widespread application with a similar number of strides [40]. Based on the work by Riva et al. [41], its reliability is 30% when calculated over 14 strides. The other investigated nonlinear indexes were selected based on the available number of strides per trial, ensuring a reliability of at least 20% [41].

For RQA calculation, the state space was constructed with an embedding dimension of dE = 5 and a time delay of 10 samples [30,34] for all the subjects and both tasks, to ensure comparability. Raw unfiltered data were analysed to assure that information was not lost or altered.

To guarantee reliability of MSE results [42], sensitivity to radius values was verified (radius = 0.10, 0.15, 0.20, 0.25, and 0.30) for each τ for the two groups and relative consistency was verified for radius values below and above the selected one.

### 2.6. Data Analysis

A Kolmogorov–Smirnov test was performed to test the normal distribution of the estimated parameters, which was not verified for all the parameters.

Estimated parameters were compared to those obtained in a previous study [28] on groups of typically developing subjects and healthy young adults.

Statistical analysis was performed to test the following:(i)Differences between DS and typically developing subjects (6–25 years old, considered as a single group—TD): Mann–Whitney U test, level of significance 0.05.(ii)Differences between DS subjects and specific age groups of typically developing subjects (6-, 7–8-, 9–10-, 15-, and 25-year-old subjects—6 YC, 7–8 YC, 9–10 YC, 15 YA, and 25 YA, respectively): Kruskal–Wallis test, level of significance 0.05. When a significant group effect was found, a multiple comparison test was performed to evaluate which of the analysed groups showed significant differences from DS. Dunn–Sidak correction was considered for post hoc analysis.

To test the correlation of the estimated indices with subject-specific motor function score, Spearman correlation coefficients ρ (significance level 0.05) were calculated between indices and (i) FIM total score, (ii) FIM motor function subscore, (iii) walking speed, and (iv) normalised walking speed.

Data and statistical analyses were performed in Matlab 2017 (The MathWorks Inc., Natick, MA, USA).

## 3. Results

For the DS group, speed and normalized speed were equal to (median [1st quartile; 3rd quartile]) 0.90 [0.75; 1.00] m/s and 56.45 [52.29; 72.20] %H/s; step width was equal to 11 [8; 13] cm; and FIM total score and motor function subscore were 87 [71; 118] and 32 [27; 35]. Individual results for each DS subject are shown in Table 3.

For each DS subject, the raw data of trunk 3D acceleration (14 complete strides) and estimated nonlinear indices are available in the Appendix A. The same information for the TD control group can be found in the Appendix A in Bisi et al., 2019 [28].

### 3.1. Differences between DS and TD Subjects

DS showed significantly lower FF (median value, 0.88) than TD (median value, 1.06); in particular, DS values were similar to those of the 15 YA and 25 YA groups and significantly lower than those of 6 YC, 7–8 YC, and 9–10 YC (median value, 1.23, 1.09, and 1.07, respectively).

HR values assessed on V and AP directions resulted lower in DS (median values: HRv, 1.33; HRap, 1.42) than in TD (median values: HRv, 2.25; HRap, 2.02): HRv and HRap in DS were significantly lower than the values obtained in each group of TD subjects. No significant difference was found for HRml.

When analysing MSE in the V and AP directions, no difference was found between DS and TD. When analysing differences between DS and specific age groups, DS showed SENv and SENap values significantly higher than 25 YA for τ = 4:6. When considering the ML direction, MSE had significantly lower results in DS than in TD for all τ values (e.g., τ = 6, DS median value 1.54, TD median value 1.65). When analysing the differences between DS and the specific age groups, DS showed the following values: lower than 6 YC for τ = 1,2,5; lower than 6 YC and 7–8 YC for τ = 1,2; for τ = 6, even if a group effect was found, DS values were not significantly different from any of the other groups.

RQA parameters results were significantly lower in DS than in TD when calculated on the V direction (DS median values: RRv 10.95, DETv 51.7, AvgLv 6.33; TD median values: RRv 13.42, DETv 77.2, AvgLv 9.94) and the AP direction (DS median values: RRap 13.09, DETap 50.0, AvgLap 7.16; TD median values: RRap 15.08, DETap 74.2, AvgLap 7.72). On the V direction, RRv in DS resulted significantly lower than in 6 YC, 9–10 YC, and 25 YA; DETv was lower than in all groups except 15 YA and AvgLv was lower than that in all the considered age groups. In the AP direction, RRap results were significantly lower in DS than in 15 YA and 25 YA; DETap was lower than in 7–8 YC, 9–10 YC, and 25 YA; and AvgLap was lower than in 25 YA. No significant difference between groups was found for RQA parameters calculated on the ML direction.

Median values and the 25th and 75th percentiles of all the analysed parameters for DS and TD, and DS and specific age groups (6 YC, 7–8 YC, 9–10 YC, 15 YA, and 25 YA) are reported in Table 4 and Table 5.

### 3.2. Correlations between FIM Scores and Nonlinear Indices

No correlation was found for FIM total score or FIM motor function subscore with FF.

FIM total score and FIM motor function showed positive correlations with HR on AP direction (total score: ρ = 0.53, *p* = 0.02; motor function subscore ρ = 0.62, *p* = 0.004). No significant correlation was found for HR calculated on the V or ML directions. 

FIM total score and FIM motor function subscore correlated positively with SEN at different values of τ on ML direction, while no significant correlation was found for MSE calculated on V and AP directions. In particular, total FIM score was positively correlated with SENml for τ = 1 ÷ 4 (0.47 < ρ < 0.51, *p* < 0.04), and FIM motor function subscore correlated positively with SENml for τ = 3, 4, 6 (0.43 < ρ < 0.53, *p* < 0.05).

As referred to RQA parameters, FIM total score correlated positively with RRap (ρ = 0.47, *p* = 0.036) and FIM motor function subscore correlated negatively with RRml (ρ = −0.50, *p* = 0.023).

### 3.3. Correlations between Walking Speed and Nonlinear Indices

No significant correlation was found for walking speed/normalised walking speed and FF.

Walking speed and normalised walking speed showed positive correlations with HR in the V direction (walking speed, ρ = 0.57, *p* = 0.008; normalised walking speed, ρ = 0.43, *p* = 0.05) and in the AP direction (walking speed: ρ = 0.56, *p* = 0.01; normalised walking speed ρ = 0.46, *p* = 0.04). No significant correlation was found for HR calculated in the ML direction.

Walking speed and normalised walking speed positively correlated with MSE in the V and ML directions. No significant correlation was found for MSE calculated in the AP direction. In particular, walking speed correlated with SEN v for τ > 4 (0.47 < ρ < 0.51, *p* < 0.04) and normalised walking speed for τ = 6 (ρ = 0.48, *p* = 0.03); walking speed correlated positively with SENml for all the values of τ (0.60 < ρ < 0.82, *p* < 0.002); and normalised walking speed positively correlated with SENml for τ = 2,3,4,6 (0.49 < ρ < 0.62, *p* < 0.03).

When considering RQA parameters, significant negative correlations were found between walking speed and/or normalised walking speed and all RQA parameters calculated on ML direction, and between walking speed and AvgL in the AP direction, while no significant correlation was found in the V direction. In particular, RRml showed correlation coefficients ρ = −0.80 (*p* = 10^−5^) and ρ = −0.66 (*p* = 0.002) with walking speed and normalised walking speed. DETml showed correlation coefficients ρ = −0.76 (*p* = 10^−5^) and ρ = −0.65 (*p* = 0.002) with walking speed and normalised walking speed. AvgLml showed correlation coefficients ρ = −0.62 (*p* = 10^−4^) and ρ = −0.49 (*p* = 0.03) with walking speed and normalised walking speed. AvgLap correlated negatively with walking speed (ρ = −0.48, *p* = 0.03), while no significant correlation was found with normalised walking speed.

Table 6 shows Spearman correlation coefficient ρ values of significant correlations.

## 4. Discussion

The aim of this work was to analyse the gait of DS subjects by means of a wearable-sensor-based approach, quantifying a set of nonlinear indices related to the control of the centre of mass during locomotion, motor control maturation, and age [28,29]. Gait abnormalities appear quite early in the natural history of DS and correlate with cognitive development [11]. A non-invasive, user-friendly method to quantify gait parameters, such as wearable sensors, may thus pave the way for precocious clinical prognostic biomarkers.

Of the quantified indices, FF values resulted significantly lower in DS subjects compared with the whole TD group. This reduction results from a broader distribution of power in the frequency spectrum of the acceleration of the centre of mass; the clinical correlate of this finding is likely the unsteady, ataxic gait of DS subjects, which differs substantially from the cyclic 3D motion pattern of the centre of mass of TD subjects.

HR also showed significantly reduced results in DS, but only in the AP and V directions, i.e., in the sagittal plane. This finding likely has a clinical correlate in the disruption of the cyclic pattern in TD, again mirroring the unsteady, ataxic characteristic of DS gait [5], which is more evident in the sagittal plane. Nonlinear indices thus allow a characterization of motor control in these specific directions.

The clinical unsteadiness of DS gait in the sagittal plane is further confirmed by the significant reduction in RQA_V and RQA_AP, i.e., the regularity of the acceleration of the centre of mass.

Among indices on the ML direction, only SEN_ML values resulted were significantly reduced in DS, demonstrating a reduction in the complexity of motor control in the ML direction. This finding suggests that the gait pattern in DS appears highly disrupted and unsteady in the sagittal plane, but grossly repetitive and less disrupted along the ML direction. These measures (i.e., reduced HRap, HRv, SENml and RQA parameters on the ML and V direction) can be assumed to instrumentally quantify a clinical feature of ataxic gait: a common strategy to reduce pluri-directional oscillations is to extend the base of support—i.e., widen the feet distance (Table 3) [6]. This compensatory postural adjustment can be assumed to be further reinforced by two biomechanical features which are typical of DS—flat foot and valgus knee [6]—which both contribute to the increased lateral reaction forces [7].

In our cohort, the abnormalities observed in the HR are highly suggestive of ataxia, as described in previous reports on population of cerebellar ataxic participants [43]. The HR measures rhythm [44], symmetry [45], or smoothness [46] of trunk acceleration patterns during walking, and is highly correlated with gait stability and fall risk [17,47,48], which are more frequent in ataxic people. HRs metrics appear as the most representative acceleration-derived markers of the loss of ability to organize a smooth and rhythmically effective gait in people with cerebellar ataxia due to cerebellar degeneration leading to inter-joint and inter-segmental incoordination. It is of note that HR appears highly reliable in identifying even gait incoordination, given that not all our participants showed a clear-cut clinical ataxia. Our group already described biomechanical strategies that people with DS adopt to stabilize gait, such as pseudo-crouch gait [6], a gait pattern that lowers the centre of mass by increasing hip, knee, and ankle flexion. The gait thus resembles the crouch gait seen in cerebral palsy, without the clinical characteristics of it (lower limb muscles spasticity, muscle retractions, femoral anteversion).

Since the original description of Dravet syndrome, ataxia was reported in 80% of patients [49], usually appearing during childhood. Later publications reported more variable frequency of ataxia, between 50 and 80%. In many cases, ataxia is transitorily worsened after status epilepticus, and increases with age. A mouse model with loss-of-function mutations in NaV1.1 channels recapitulates all the clinical features of DS: severe drug resistant epilepsy, cognitive and behavioural impairment, and ataxia. Some experimental studies hypothesized that a decreased excitability of inhibitory cerebellar Purkinje neurons with loss of function in Nav1.1 could be responsible for clinical signs, including ataxia and reduced motor coordination [50].

Further supporting the clinical significance of the devised nonlinear indices, they significantly correlated with functional scores (i.e., FIM and walking speed), confirming their reliability in capturing clinically meaningful signatures of gait; DS subjects with higher HR in the AP and V directions and with higher SEN_ML (i.e., closer to those of TD subjects) had a better global FIM score, a high measure of independence, as well as FIM motor subscores, and walked at higher speed. The ability of the subject to perform a steadier gait is detected by FIM as a higher level of independence.

A limitation of our study is the relatively small sample size of people with DS; however, the inherent rarity of the disease (incidence ranging from 1:20,000 to 1:40,900 births [2]) makes our sample clinically meaningful.

## 5. Conclusions

The proposed nonlinear indices show a specific characterization of DS gait in the analysed group, clinically described as an ataxic gait. Ataxia entails a deficient motor coordination and control, an unsteady gait, and a tendency to stumble, among other signs. These clinical features are accurately picked up by the nonlinear indices we proposed.

The results of this study support the effectiveness of the proposed sensor-based method in detecting clinical features of DS gait. This is a promising finding, potentially providing biomarkers through a non-invasive and ecological assessment and follow up of gait abnormalities in DS.

## Figures and Tables

**Figure 1 sensors-22-02140-f001:**
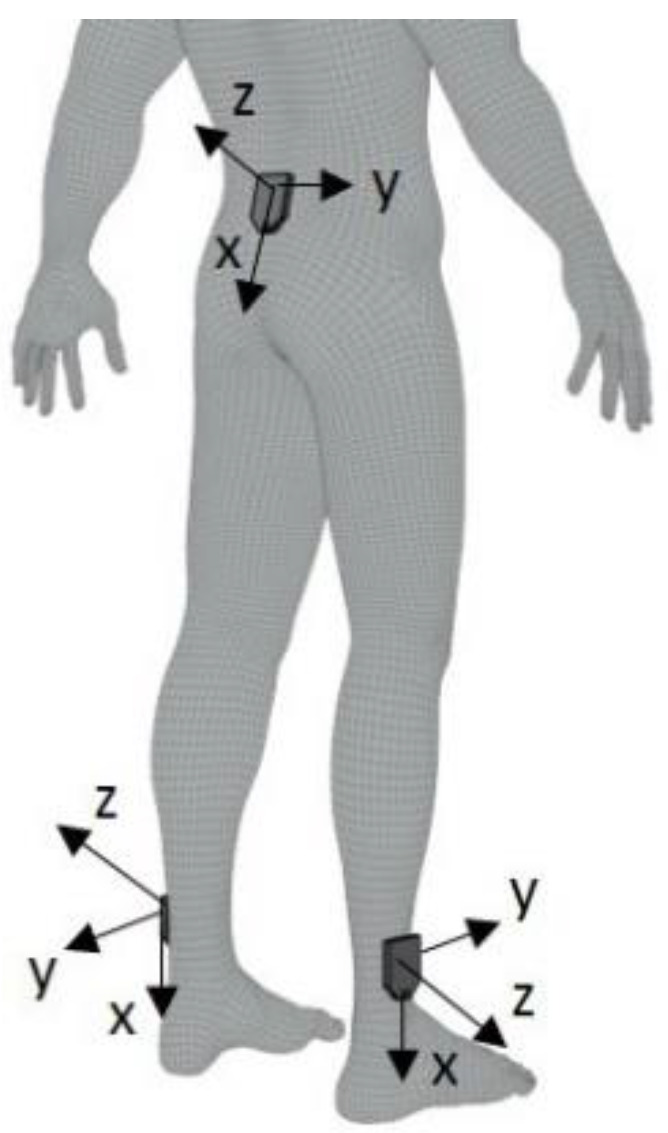
IMU placements on the different body locations and relative axis orientations.

**Table 1 sensors-22-02140-t001:** Demographic and clinical characteristics of the included DS subjects.

Subject ID	Female/Male	Age (Years)	Height (H—cm)	Body Mass (kg)
P01	F	33	147.5	52.0
P05	F	16	159.0	44.5
P08	F	14	151.0	46.0
P11	M	17	161.0	65.0
P14	M	15	151.0	47.5
P15	F	23	124.0	55.0
P17	M	20	164.0	80.0
P18	M	23	194.5	79.5
P24	F	11	135.0	30.5
P33	M	9	136.5	36.0
P35	F	28	169.5	53.5
P36	F	28	173.5	54.5
P41	F	19	161.0	72.0
P42	F	17	150.5	40.0
P43	M	14	156.5	45.0
P44	M	9	136.0	28.5
P45	F	18	139.0	42.0
P52	F	29	152.0	50.0
P54	F	12	148.0	29.0
P55	F	10	134.5	36.0
P59	F	19	162.0	62.5
**Median**	-	17	151	48
1st quartile	-	14	141	41
3rd quartile	-	22	161	55
**Mean**	-	18	153	50
Std	-	7	16	15

**Table 2 sensors-22-02140-t002:** Details of TD age groups participating in the study. Data were presented in Bisi et al. [28].

Abbreviation	Description	Female/Male	Age (Years)	Height (cm)	Body Mass (kg)
6 YC	16 6-year-old children	8 F/8 M	6 ± 0	119 ± 4	23 ± 2
7 YC	16 7-year-old children	8 F/8 M	7 ± 0	127 ± 5	29 ± 5
8 YC	16 8-year-old children	8 F/8 M	8 ± 0	130 ± 5	29 ± 6
9 YC	16 9-year-old children	8 F/8 M	9 ± 0	138 ± 6	34 ± 6
10 YC	16 10-year-old children	8 F/8 M	10 ± 0	141 ± 5	37 ± 5
15 YA	16 15-year-old adolescents	8 F/8 M	15 ± 0	168 ± 9	60 ± 13
25 YA	16 25-year-old adults	8 F/8 M	25 ± 1	171 ± 9	64 ± 11

**Table 3 sensors-22-02140-t003:** Walking speed, normalised walking speed, and functional independence measure (total score and motor function subscore) obtained for DS subjects.

Subject ID	Speed (m/s)	Normalized Speed (%H/s)	Step Width (cm)	FIM Total Score	FIM Motor Function Subscore
P01	0.90	62.88	16	63	26
P05	0.80	53.44	8	104	30
P08	N/A	N/A	N/A	30	13
P11	0.50	29.30	12	71	30
P14	1.00	68.96	4	123	35
P15	0.80	62.13	12	35	12
P17	0.90	56.45	9	119	35
P18	1.50	76.19	11	111	35
P24	1.30	93.71	17	122	35
P33	0.60	42.27	11	77	25
P35	0.90	52.99	15	70	26
P36	0.90	52.75	10	89	29
P41	0.70	40.48	11	118	34
P42	1.10	77.02	6	120	35
P43	1.30	84.25	13	81	27
P44	0.70	51.83	5	84	34
P45	0.60	38.29	15	84	28
P52	0.90	58.99	7	123	35
P54	0.80	53.41	8	115	35
P55	1.00	75.44	10	57	35
**Median**	0.90	56.45	11	87	32
1st quartile	0.75	52.29	8	71	27
3rd quartile	1.00	72.20	13	118	35
**Mean**	0.91	59.51	11	90	30
**Std**	0.26	16.63	4	29	7

**Table 4 sensors-22-02140-t004:** Median (25th and 75th percentile) values of estimated nonlinear parameters for DS and TD. Asterisks indicate significant differences.

		DS	TD	
**FF**		0.88 [0.82 0.96]	1.06 [0.97 1.14]	* DS < TD
**HRv**		1.33 [1.13 1.53]	2.25 [1.92 2.74]	* DS < TD
**HRml**		1.27 [1.18 1.45]	1.41 [1.20 1.74]	
**HRap**		1.42 [1.13 1.49]	2.02 [1.72 2.42]	* DS < TD
**SEN V**	**tao = 1**	0.37 [0.31 0.42]	0.39 [0.35 0.44]	
	**tao = 2**	0.62 [0.49 0.73]	0.60 [0.52 0.69]	
	**tao = 3**	0.85 [0.63 0.98]	0.78 [0.69 0.91]	
	**tao = 4**	1.07 [0.76 1.20]	0.92 [0.82 1.06]	
	**tao = 5**	1.20 [0.88 1.36]	1.05 [0.92 1.22]	
	**tao = 6**	1.23 [1.03 1.49]	1.13 [0.97 1.28]	
**SEN ML**	**tao = 1**	0.41 [0.39 0.51]	0.51 [0.47 0.55]	* DS < TD
	**tao = 2**	0.71 [0.64 0.85]	0.86 [0.79 0.95]	* DS < TD
	**tao = 3**	0.97 [0.83 1.16]	1.15 [1.04 1.30]	* DS < TD
	**tao = 4**	1.16 [1.00 1.44]	1.42 [1.26 1.54]	* DS < TD
	**tao = 5**	1.40 [1.15 1.69]	1.57 [1.42 1.75]	* DS < TD
	**tao = 6**	1.54 [1.26 1.77]	1.65 [1.46 1.85]	
**SEN AP**	**tao = 1**	0.40 [0.32 0.52]	0.40 [0.34 0.44]	
	**tao = 2**	0.63 [0.51 0.80]	0.60 [0.53 0.67]	
	**tao = 3**	0.74 [0.67 0.99]	0.76 [0.68 0.86]	
	**tao = 4**	0.94 [0.84 1.13]	0.88 [0.78 1.00]	
	**tao = 5**	1.09 [0.92 1.29]	1.00 [0.87 1.13]	
	**tao = 6**	1.22 [1.02 1.29]	1.06 [0.94 1.24]	
**RQA V**	**RR**	10.95 [9.13 13.83]	13.42 [12.10 15.11]	* DS < TD
	**DET**	51.68 [37.15 61.82]	77.23 [67.39 82.59]	* DS < TD
	**AvgL**	6.33 [5.42 7.34]	9.94 [8.41 11.77]	* DS < TD
**RQA ML**	**RR**	8.22 [7.76 8.71]	7.98 [7.64 8.65]	
	**DET**	47.25 [36.81 64.97]	44.11 [37.77 49.77]	
	**AvgL**	6.11 [5.42 7.44]	5.97 [5.72 6.49]	
**RQA AP**	**RR**	13.09 [11.73 15.01]	15.08 [13.54 16.29]	* DS < TD
	**DET**	50.01 [44.36 64.93]	74.17 [62.40 80.65]	* DS < TD
	**AvgL**	7.16 [6.29 9.30]	7.72 [6.97 8.18]	

**Table 5 sensors-22-02140-t005:** Median (25th and 75th percentile) values of estimated nonlinear parameters for DS, TD, and specific TD age groups (6 YC, 7–8 YC, 9–10 YC, 15 YA, and 25 YA). Asterisks indicate significant differences, described between brackets.

		DS	6 YC	7–8 YC	9–10 YC	15 YA	25 YA	
**FF**		0.88 [0.82 0.96]	1.23 [1.16 1.35]	1.09 [1.03 1.15]	1.07 [1.01 1.11]	0.98 [0.91 1.00]	0.89 [0.85 0.95]	* (DS < 6 YC; DS < 7–8 YC; DS < 9–10 YC)
**HRv**		1.33 [1.13 1.53]	2.15 [1.55 2.59]	2.21 [1.92 2.61]	2.41 [2.12 3.03]	1.86 [1.69 2.48]	2.31 [2.04 2.70]	* (DS < 6 YC; DS < 7–8 YC; DS < 9–10 YC; DS < 15 YA; DS < 25 YA)
**HRml**		1.27 [1.18 1.45]	1.45 [1.34 1.72]	1.38 [1.19 1.81]	1.51 [1.19 1.89]	1.22 [1.12 1.64]	1.42 [1.24 1.55]	
**HRap**		1.42 [1.13 1.49]	1.95 [1.53 2.51]	1.97 [1.77 2.31]	2.16 [1.81 2.61]	1.76 [1.62 2.29]	2.19 [1.74 2.44]	* (DS < 6 YC; DS < 7.8 YC; DS < 9–10 YC; DS < 15 YA; DS < 25 YA)
**SEN V**	**tao = 1**	0.37 [0.31 0.42]	0.42 [0.38 0.50]	0.41 [0.37 0.45]	0.39 [0.34 0.45]	0.39 [0.35 0.43]	0.33 [0.29 0.38]	
	**tao = 2**	0.62 [0.49 0.73]	0.63 [0.58 0.70]	0.60 [0.56 0.71]	0.59 [0.52 0.71]	0.63 [0.58 0.69]	0.50 [0.44 0.56]	
	**tao = 3**	0.85 [0.63 0.98]	0.83 [0.74 0.91]	0.80 [0.72 0.95]	0.78 [0.69 0.93]	0.82 [0.72 0.89]	0.65 [0.59 0.70]	
	**tao = 4**	1.07 [0.76 1.20]	0.97 [0.84 1.10]	1.00 [0.88 1.07]	0.92 [0.87 1.06]	1.05 [0.84 1.07]	0.76 [0.67 0.81]	* (DS > 25 YA)
	**tao = 5**	1.20 [0.88 1.36]	1.13 [0.89 1.24]	1.13 [0.98 1.22]	1.06 [0.99 1.20]	1.05 [0.97 1.25]	0.83 [0.74 0.97]	* (DS > 25 YA)
	**tao = 6**	1.23 [1.03 1.49]	1.04 [0.92 1.25]	1.13 [1.07 1.29]	1.20 [1.02 1.33]	1.16 [0.98 1.35]	0.93 [0.79 1.06]	* (DS > 25 YA)
**SEN ML**	**tao = 1**	0.41 [0.39 0.51]	0.58 [0.52 0.62]	0.50 [0.48 0.55]	0.52 [0.47 0.54]	0.47 [0.45 0.54]	0.51 [0.46 0.54]	* (DS < 6 YC)
	**tao = 2**	0.71 [0.64 0.85]	0.98 [0.90 1.05]	0.86 [0.80 0.94]	0.86 [0.79 0.92]	0.83 [0.78 0.91]	0.83 [0.74 0.86]	* (DS < 6 YC)
	**tao = 3**	0.97 [0.83 1.16]	1.35 [1.23 1.41]	1.17 [1.09 1.30]	1.14 [1.05 1.25]	1.11 [0.99 1.22]	1.08 [0.87 1.14]	* (DS < 6 YC; DS < 7.8 YC)
	**tao = 4**	1.16 [1.00 1.44]	1.53 [1.49 1.72]	1.45 [1.28 1.61]	1.41 [1.23 1.51]	1.36 [1.23 1.52]	1.28 [1.06 1.42]	* (DS < 6 YC; DS < 7.8 YC)
	**tao = 5**	1.40 [1.15 1.69]	1.76 [1.51 1.92]	1.64 [1.46 1.80]	1.55 [1.42 1.63]	1.58 [1.41 1.75]	1.41 [1.14 1.54]	* (DS < 6 YC)
	**tao = 6**	1.54 [1.26 1.77]	1.72 [1.45 2.05]	1.72 [1.58 1.91]	1.63 [1.49 1.86]	1.58 [1.40 1.74]	1.48 [1.23 1.70]	
**SEN AP**	**tao = 1**	0.40 [0.32 0.52]	0.43 [0.37 0.48]	0.40 [0.36 0.42]	0.41 [0.35 0.45]	0.35 [0.32 0.38]	0.39 [0.34 0.47]	
	**tao = 2**	0.63 [0.51 0.80]	0.66 [0.55 0.79]	0.61 [0.54 0.66]	0.61 [0.54 0.68]	0.56 [0.50 0.60]	0.58 [0.52 0.65]	
	**tao = 3**	0.74 [0.67 0.99]	0.90 [0.70 1.02]	0.79 [0.70 0.85]	0.78 [0.69 0.85]	0.70 [0.61 0.77]	0.71 [0.65 0.81]	
	**tao = 4**	0.94 [0.84 1.13]	1.00 [0.83 1.17]	0.90 [0.82 1.00]	0.88 [0.79 0.99]	0.82 [0.72 0.93]	0.79 [0.72 0.96]	* (DS > 25 YA)
	**tao = 5**	1.09 [0.92 1.29]	1.15 [1.03 1.31]	1.01 [0.93 1.13]	0.97 [0.86 1.12]	0.94 [0.83 1.06]	0.86 [0.75 0.91]	* (DS > 25 YA)
	**tao = 6**	1.22 [1.02 1.29]	1.26 [1.06 1.56]	1.12 [1.01 1.22]	1.06 [0.95 1.23]	1.01 [0.86 1.12]	0.88 [0.76 0.99]	* (DS > 25 YA)
**RQA V**	**RR**	10.95 [9.13 13.83]	14.22 [11.86 16.83]	13.28 [11.72 14.45]	13.39 [12.63 15.07]	11.91 [11.37 15.01]	14.27 [13.60 15.37]	* (DS < 6 YC; DS < 9–10 YC; DS < 25 YA)
	**DET**	51.68 [37.15 61.82]	75.07 [63.84 83.17]	75.86 [64.07 80.35]	76.07 [67.21 81.03]	73.71 [66.63 80.22]	84.43 [80.73 87.30]	* (DS < 6 YC; DS < 7.8 YC; DS < 9–10 YC; DS < 25 YA)
	**AvgL**	6.33 [5.42 7.34]	9.68 [8.03 11.85]	9.33 [8.54 10.97]	10.32 [8.68 11.74]	9.05 [7.68 11.02]	11.72 [10.39 12.67]	* (DS < 6 YC; DS < 7.8 YC; DS < 9–10 YC; DS < 15 YA; DS < 25 YA)
**RQA ML**	**RR**	8.22 [7.76 8.71]	7.83 [7.64 8.94]	7.86 [7.59 8.34]	7.98 [7.65 8.55]	8.30 [7.88 8.63]	8.69 [7.67 9.52]	
	**DET**	47.25 [36.81 64.97]	40.42 [30.66 45.75]	44.11 [37.80 49.55]	41.75 [36.65 45.76]	45.21 [41.58 50.23]	54.68 [48.90 59.50]	*
	**AvgL**	6.11 [5.42 7.44]	5.80 [5.60 6.14]	5.84 [5.72 6.19]	5.99 [5.81 6.40]	6.16 [5.81 6.50]	6.86 [6.24 7.20]	*
**RQA AP**	**RR**	13.09 [11.73 15.01]	12.50 [11.15 15.23]	15.03 [14.09 16.18]	15.01 [13.20 15.93]	15.82 [14.36 16.41]	16.59 [15.14 18.00]	* (DS < 15 YA; DS < 25 YA)
	**DET**	50.01 [44.36 64.93]	52.31 [46.66 71.98]	75.43 [64.53 80.67]	76.47 [66.47 81.31]	64.34 [57.21 75.33]	83.54 [74.20 89.24]	* (DS < 7.8 YC; DS < 9–10 YC; DS < 25 YA)
	**AvgL**	7.16 [6.29 9.30]	6.66 [6.05 7.49]	7.68 [7.07 8.00]	7.78 [7.07 8.11]	7.47 [6.85 8.36]	9.75 [8.14 10.81]	* (DS < 25 YA)

**Table 6 sensors-22-02140-t006:** Spearman correlation coefficients ρ for indices and (i) FIM total score and (ii) FIM motor function subscore; (iii) walking speed and (iv) normalized walking speed.

		FIM Total Score	FIM Motor Function Subscore	Speed (m/s)	Normalized Speed (%H/s)
**FF**		--	--	--	--
**HRv**		--	--	0.57	0.43
**HRml**		--	--	--	--
**HRap**		0.62	0.53	0.56	0.46
**SEN V**	**tao = 1**	--	--	--	--
	**tao = 2**	--	--	--	--
	**tao = 3**	--	--	--	--
	**tao = 4**	--	--	0.47	--
	**tao = 5**	--	--	0.51	--
	**tao = 6**	--	--	0.49	0.48
**SEN ML**	**tao = 1**	0.49	--	0.6	--
	**tao = 2**	0.51	--	0.72	0.49
	**tao = 3**	0.5	0.45	0.81	0.59
	**tao = 4**	0.47	0.43	0.82	0.62
	**tao = 5**	--	--	0.65	--
	**tao = 6**	--	0.53	0.76	0.54
**SEN AP**	**tao = 1**	--	--	--	--
	**tao = 2**	--	--	--	--
	**tao = 3**	--	--	--	--
	**tao = 4**	--	--	--	--
	**tao = 5**	--	--	--	--
	**tao = 6**	--	--	--	--
**RQA V**	**RR**	--	--	--	--
	**DET**	--	--	--	--
	**AvgL**	--	--	--	--
**RQA ML**	**RR**	--	−0.5	−0.8	−0.66
	**DET**	--	--	−0.76	−0.65
	**AvgL**	--	--	−0.62	−0.49
**RQA AP**	**RR**	0.47	--	--	--
	**DET**	--	--	--	--
	**AvgL**	--	--	−0.48	--

## Data Availability

Data of DS subjects may be available upon requested by registered researchers for demonstrated scientific purposes. Data of TD subjects are published as Appendix A.

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
