# Peer review of "Quantitative Characterization of Motor Control during Gait in Dravet Syndrome Using Wearable Sensors: A Preliminary Study"

_sensors, 2022, doi:10.3390/s22062140_

Round 1

Reviewer 1 Report

Thank you for submitting this paper to Sensors. The manuscript under consideration: "Quantitative characterization of motor control during gait in Dravet Syndrome using wearable sensors: a preliminary study" is an interesting article on an important topic in Sensors. However, there are a few concerns.

・In the study  patients with DS and healthy control participated. How did the authors determine the sample appropriate size?

・How did you eliminate the impact of growth?

・Control group were physique matched?

 ãƒ»Why didn't you survey other physical function (ex; six-minute walk test)?

・Give more information on the age of the sample, why it was chosen, and give some context for results of neurologic examination at this age.

・What is the biomechanics and physiological basis for ataxic characteristic of DS gait, more evident in the sagittal plane? Discussion is biomechanics and physiological explanation is inadequate.

Reviewer 2 Report

This manuscript is well-written overall. Below is my comments.

In the introduction, please add hypothesis.

In the methods, please justify why these measures were selected. Additionally, please justify why the authors compared to age groups of typically developing subjets.

Table 3 is difficult to read. Please consider to split the table into two or three.

Throughout tables, please consider adding mean and standard deviations in addition to what is reported now even if the data distribution is not normal distribution. I think mean and standard deviation is sometimes easier to catch.

Round 2

Reviewer 1 Report

I think the manuscript is sufficiently improved.

Author Response

Thank you for your comment